# Glia Not Neurons: Uncovering Brain Dysmaturation in a Rat Model of Alzheimer’s Disease

**DOI:** 10.3390/biomedicines9070823

**Published:** 2021-07-15

**Authors:** Ekaterina A. Rudnitskaya, Tatiana A. Kozlova, Alena O. Burnyasheva, Natalia A. Stefanova, Nataliya G. Kolosova

**Affiliations:** Institute of Cytology and Genetics, Siberian Branch of Russian Academy of Sciences (SB RAS), 10 Lavrentyeva Ave., 630090 Novosibirsk, Russia; kozlova@bionet.nsc.ru (T.A.K.); burnyasheva@bionet.nsc.ru (A.O.B.); stefanovan@bionet.nsc.ru (N.A.S.); kolosova@bionet.nsc.ru (N.G.K.)

**Keywords:** neurogenesis, neuron, glia, postnatal development, hippocampus, prefrontal cortex, Alzheimer’s disease, OXYS rats

## Abstract

Sporadic Alzheimer’s disease (AD) is a severe disorder of unknown etiology with no definite time frame of onset. Recent studies suggest that middle age is a critical period for the relevant pathological processes of AD. Nonetheless, sufficient data have accumulated supporting the hypothesis of “neurodevelopmental origin of neurodegenerative disorders”: prerequisites for neurodegeneration may occur during early brain development. Therefore, we investigated the development of the most AD-affected brain structures (hippocampus and prefrontal cortex) using an immunohistochemical approach in senescence-accelerated OXYS rats, which are considered a suitable model of the most common—sporadic—type of AD. We noticed an additional peak of neurogenesis, which coincides in time with the peak of apoptosis in the hippocampus of OXYS rats on postnatal day three. Besides, we showed signs of delayed migration of neurons to the prefrontal cortex as well as disturbances in astrocytic and microglial support of the hippocampus and prefrontal cortex during the first postnatal week. Altogether, our results point to dysmaturation during early development of the brain—especially insufficient glial support—as a possible “first hit” leading to neurodegenerative processes and AD pathology manifestation later in life.

## 1. Introduction

The most common type of Alzheimer’s disease (AD) (∼95% of cases) is sporadic AD, which is a progressive neurodegenerative disorder of middle-aged-to-old individuals. The pathological changes associated with AD are thought to begin many years before the emergence of clinical symptoms. Accumulating data indicate that middle age is a critical period for the relevant pathological processes. Moreover, it is theorized that the prerequisites for subsequent development of neurodegenerative disorders are formed during completion of brain maturation [1,2,3]: the so-called neurodevelopmental hypothesis of neurodegenerative disorders [4,5]. The third trimester of pregnancy in humans is a crucial period when proper cytoarchitecture of the brain and functional connections among neurons are formed. Additionally, the peak of gliogenesis corresponds to the third trimester; moreover, the blood–brain barrier arises, and the immune system develops and consolidates at this time [6,7,8]. Disturbances of brain development in the third trimester of pregnancy may cause cognitive and behavioral disorders, e.g., encephalopathy, or fetal alcohol syndrome [9,10]. Nevertheless, to date, little is known about the long-lasting effects of developmental alterations on adult brain function and about a possible contribution and role of these alterations in the development of the neurodegenerative processes that lead to such conditions as AD, Parkinson’s disease, or others. Thus, the first step in the investigation of these effects may be made by studying the early phenotype of model organisms of neurodegenerative disorders.

Although neurons are the most characteristic cells of the nervous system and are primarily responsible for information transmission, they are dependent on, interact with, and are surrounded by glial cells [6]. The two main glial subsets in the central nervous system are macroglia, including astrocytes and oligodendrocytes, and microglia. Although astrocytes and microglia are fundamentally different in origin and function, they affect the same developmental processes: neuronal survival, synaptic development and remodeling, axonal development and guidance, neural-circuit formation, oligodendrocyte differentiation and myelination, angiogenesis, and vascularization [10,11,12]. Furthermore, astrocytes take part in the construction of the blood–brain barrier [13]. Due to their important role in brain development, dysfunction of astrocytes or microglia in this period may contribute to neurodevelopmental disorders and potentially even late-onset neuropathology [14].

To study the long-lasting consequences of early alterations of brain development and a possible contribution of these alterations to the pathogenesis of neurodegenerative processes, animal models can be used. It is generally accepted that the developmental events occurring in the third trimester of pregnancy in humans match those taking place from birth to postnatal day 7 (PND7) in rodents [15]. The first postnatal week appears to be a critical period for the maturation of prefrontal–hippocampal networks in rodents: directed prefrontal–hippocampal communication is initiated at PND3 and lasts until the beginning of the second postnatal week [16,17]. Prefrontal–hippocampal networks are important for associative learning in adult animals [18]. Thus, alterations of these networks’ maturation during the critical developmental period may result in altered associative learning in adulthood.

Previously, we have reported that senescence-accelerated OXYS rats may be regarded as a suitable model of the most common, sporadic, type of AD [19]. Indeed, already at 3 months of age, OXYS rats demonstrate the first signs of neurodegeneration: neuronal cell death, synaptic dysfunction, hyperphosphorylation of tau protein, and mitochondrial dysfunction, all of which together lead to behavioral alterations and memory deterioration [20]. By 18 months of age, the neurodegenerative processes intensify against the background of amyloid β (Aβ) accumulation and the formation of Aβ plaques [21]. More recently [22,23], we showed that the duration of gestation is shorter in OXYS rats than in a control strain (Wistar rats); furthermore, we observed signs of retardation of brain development in the second decade of life (from PND10 to PND20). Current work continues our previous researches of early development of OXYS rats. We hypothesized that the developmental features occurring during the period critical for the maturation of prefrontal–hippocampal networks (i.e., the first postnatal week) may contribute to the neurodegenerative processes and behavioral alterations taking place late in life in OXYS rats. To test the hypothesis, we examined neuronal and glial-cell density as well as some functional parameters of cells in the prefrontal cortex (PFC) and hippocampus of OXYS rats from birth to PND7. However, future studies are needed to demonstrate the direct link between the early phenotype and age-related neurodegenerative changes in OXYS rats.

## 2. Materials and Methods

### 2.1. Animals

The OXYS rat strain was developed at the Institute of Cytology and Genetics (ICG), SB RAS (Novosibirsk, Russia), from a Wistar stock as described earlier [19]. This senescence-accelerated strain of age-matched male Wistar rats was obtained from the Breeding Experimental Animal Laboratory of the ICG SB RAS (Novosibirsk, Russia). The animals were kept under standard laboratory conditions (22 ± 2 °C, 60% relative humidity, and 12 h light/12 h dark cycle) and had ad libitum access to standard rodent feed (PK-120-1, Laboratorsnab, Ltd., Moscow, Russia) and water.

### 2.2. Examination of Body and Brain Weights

We assessed the body weight, brain weight, and brain-to-body weight ratio [meaning (brain weight ÷ body weight) × 100%] of male pups of OXYS and Wistar strains on PND1, PND3, PND5, and PND7 (n = 8 to 10 per strain and age).

### 2.3. Tissue Preparation

Male pups of OXYS and Wistar strains were decapitated at PND0, PND1, PND3, PND5, and PND7; the brains were carefully excised, and the hemispheres were separated and immediately fixed in 4% paraformaldehyde in phosphate-buffered saline (PBS) at room temperature (RT) for 48 h, followed by cryoprotection in 30% sucrose in PBS at 4 °C for 48 h. Then, the brains were frozen and stored at −70 °C until further processing.

### 2.4. Immunohistochemistry

Brain sagittal sections (20 μm thick) of OXYS and Wistar rats (n = 4 to 6 per strain and age) were prepared on a Microm HM-505 N cryostat (Microm, Walldorf, Germany) at −20 °C and transferred onto polysine-glass slides (Menzel-Glaser, Braunschweig, Germany). After serial washes with PBS, the slices were incubated at RT for 15 min in PBS-plus (PBS with 0.1% of Triton X-100) and for 1 h in 3% bovine serum albumin (BSA; cat. # A3294, Sigma-Aldrich, St. Louis, MO, USA) in PBS to permeabilize the tissues and to block nonspecific binding sites and then were incubated overnight with primary antibodies at 4 °C. The primary antibodies were all diluted 1:250 with 3% BSA in PBS; these were antibodies to Ki67, nestin, vimentin, glial fibrillary acid protein (GFAP), doublecortin (DCX), Fox-3 (NeuN), Iba1, and CD68 (cat. ## ab15580, ab6142, ab24525, ab7260, ab54739, ab177487, ab5076, and ab31630, respectively, Abcam, Cambridge, MA, USA). After several washes with PBS, the slices were probed with secondary antibodies conjugated with Alexa Fluor 488, 568, or 555 (cat. ## ab150073, ab175472, and ab150170, respectively, Abcam) in PBS (1:250) for 1 h at RT and next were washed in PBS. The slices were coverslipped with the Fluoroshield mounting medium containing 4′,6-diamidino-2-phenylindole (DAPI; cat. # ab104139, Abcam). Negative controls were processed in an identical manner except that a primary antibody was not included. The Ki67, nestin, vimentin, GFAP, DCX, NeuN, Iba1, and CD68 signals were detected under a microscope with a 40× objective lens (Axioskop 2 plus, Zeiss, Oberkochen, Germany). The microscopy was conducted at the Multi-Access Center for Microscopy of Biological Objects (ICG SB RAS, Novosibirsk, Russia). Identification of brain structures (PFC and CA1, CA3, and dentate gyrus [DG] regions of the hippocampus) was performed according to Paxinos and Watson (Lateral 0.40 to Lateral 0.90 mm) [24]. Identification of cell types was carried out according to protein markers described by Encinas and colleagues [25].

### 2.5. Calculation of Cell Density and Other Parameters

Total numbers of proliferating (Ki67-positive) cells, quiescent (nestin-positive and vimentin-positive) neural progenitors (QNPs), amplifying (nestin-positive) neural progenitors (ANPs), neuroblasts (DCX-positive), immature (DCX-positive and NeuN-positive) and mature (NeuN-positive) neurons as well as astrocyte progenitors (vimentin-positive and GFAP-positive) and astrocytes (GFAP-positive), resting (Iba1-positive) and activated (Iba1-positive and CD68-positive) microglia were determined by means of the ZEN software (Zeiss). To evaluate the density of proliferating cells, QNPs, and ANPs, the total number of counted cells was divided by DG area, then averaged in each group of 2–3 slices per animal, and presented as the number of cells per 1 mm^2^. To assess the density of neuroblasts, immature and mature neurons, astrocytes and their progenitors as well as resting and activated microglia, the total number of counted cells was divided by the area of the hippocampus and PFC, then averaged in each group of 2–3 slices per animal, and presented as the number of cells per 1 mm^2^. To evaluate the sizes of neuronal nuclei, we measured and averaged the diameters of 40 randomly selected neurons’ nuclei per section for the hippocampus (neurons were chosen from pyramidal layers of CA1 and CA3 regions and from the granular layer of the DG) and the PFC, then computed the size of the nuclei via the formula (nuclear size = π × (nuclear diameter/2)^2^). To determine the number of radial glial-cell processes, we counted radially oriented vimentin-positive processes that permeated all cortical layers of the PFC. The area of microglia in molecular layers of DG, CA1, and CA3 regions of the hippocampus as well as in cortical layer I was calculated in the ImageJ software (NIH, Bethesda, MD, USA). To determine the percentage of dying cells consumed by microglia, we counted all pyknotic nuclei and the pyknotic nuclei covered by microglial cytoplasm.

### 2.6. A Terminal Deoxynucleotidyl Transferase-Mediated Deoxyuridine Triphosphate Nick End Labeling (TUNEL) Assay

Apoptosis was analyzed by a TUNEL assay by means of a DeadEnd Fluorometric TUNEL System (cat. # G3250, Promega, Madison, WI, USA). Tissue slices were coverslipped with the Fluoroshield mounting medium containing DAPI (cat. # ab104139, Abcam). TUNEL signals were counted under the microscope with the 40× objective lens (Axioskop 2 plus, Zeiss) and then averaged in each group of three slices per animal.

### 2.7. Statistics

The data were subjected to two-way analysis of variance (ANOVA) in Statistica 8.0 software (TIBCO Software Inc., Palo Alto, CA, USA). The genotype (strain) and age were chosen as independent variables. The Newman–Keuls post hoc test was applied to significant main effects and interactions in order to assess differences between some sets of means. The data are presented as mean ± standard error of the mean (SEM). The differences were considered statistically significant at *p* < 0.05.

## 3. Results

### 3.1. Body and Brain Weights and the Brain-to-Body Weight Ratio in OXYS and Wistar Pups

The ANOVA revealed that body and brain weights naturally increased with age in both rat strains (F_3,99_ = 130.6, *p* < 0.0001 and F_3,99_ = 424.4, *p* < 0.0001, respectively; Table 1). Additionally, both parameters were lower in OXYS rats (F_1,99_ = 23.5, *p* < 0.0001 for body weight and F_1,99_ = 21.0, *p* < 0.0001 for brain weight). The brain-to-body weight ratio was affected by the genotype (F_1,99_ = 8.0, *p* = 0.006) and age (F_3,99_ = 5.9, *p* < 0.001), and there was an interaction between these factors (F_3,99_ = 6.1, *p* < 0.001). In Wistar rats, the peak of the brain-to-body weight ratio occurred on PND3, after which this parameter started to decrease because of the rapid growth of the body. By contrast, in OXYS rats, the brain-to-body weight ratio increased until PND5, thus becoming higher than that in Wistar rats at this age (*p* < 0.001). The reversed U shape of the brain-to-body weight ratio may be explained by differences in body and brain weight changes: while body weight went up exponentially (R^2^ = 0.9944 for Wistar rats and R^2^ = 0.9886 for OXYS rats), brain weight increased linearly (R^2^ = 0.992 for Wistar rats and R^2^ = 0.9991 for OXYS rats).

### 3.2. Neurogenesis in the DG of Neonatal OXYS and Wistar Pups

We analyzed the density of proliferating Ki67-positive cells (Figure 1a,c). ANOVA revealed that this parameter was affected by age of the animals (F_4,66_ = 6.1, *p <* 0.001). The peak of proliferating-cell density in the DG of the hippocampus occurred at PND1 in pups of both rat strains. In Wistar rats, the parameter slightly declined until PND7, whereas in OXYS rats, it decreased significantly from PND1 to PND3 (*p* < 0.001) and then continued to go down until PND7.

Next, we investigated progenitor cell density in the DG of OXYS and Wistar rats (Figure 1b,d). ANOVA revealed that the density of QNPs, which may develop into a neuronal or glial cell lineage, decreased with age (F_4,89_ = 5.9, *p* < 0.001), with a peak at PND1 in both rat strains. Conversely, the density of ANPs, which give rise to the neuronal cell lineage, increased with age (F_4,87_ = 12.6, *p* < 0.0001), reaching a maximum at PND7 in Wistar rats, consistently with literature data about postnatal neurogenesis in rodents [26]. Furthermore, there were two peaks of ANP density in the DG of OXYS rats: at PND3 and PND7. It is important to point out that OXYS pups were born with a five-fold decrease in ANP density in the DG relative to Wistar rats (*p* < 0.001); however, by PND1, this difference became insignificant (*p* = 0.27).

### 3.3. Neuronal-Cell Density in the Hippocampus and PFC of Neonatal OXYS and Wistar Pups

Next, we analyzed the density of the neuronal-cell lineage (i.e., neuroblasts and immature and mature neurons) in the hippocampus (Figure 2a,c) and PFC (Figure 2b,d) of OXYS and Wistar pups. In the hippocampus, the density of neuroblasts was influenced by age (F_4,49_ = 68.9, *p* < 0.0001), and there was an interaction between factors “age” and “genotype” (F_4,49_ = 2.8, *p* = 0.034). In Wistar rats, the density of neuroblasts rose from PND1 to PND3 (*p* < 0.034) with a five-fold diminution by PND5 (*p* < 0.0001) and a subsequent twofold increase by PND7 (*p* = 0.014). At birth, the density of neuroblasts was higher in OXYS rats than in Wistar rats (*p* = 0.018); the parameter decreased by PND1 (*p* = 0.018) reaching the level of Wistar rats. The density of neuroblasts continued to go down from PND3 to PND5 (*p* < 0.0001) and then went up until PND7 (*p* < 0.006) in the hippocampus of OXYS rats. The density of immature neurons decreased with age (F_4,47_ = 83.3, *p* < 0.0001). By contrast, the parameter increased from PND5 to PND7 in both rat strains (*p* = 0.010 for Wistar rats and *p* = 0.006 for OXYS rats). We did not notice any inter-strain differences in immature neuron density in the hippocampus. As for mature neuron density, we documented the effects of age (F_4,50_ = 16.4, *p* < 0.0001) and genotype (F_1,50_ = 5.6, *p* = 0.021) on this parameter. Indeed, in Wistar rats, the density of mature neurons increased from PND0 to PND1 (*p* = 0.003), decreased twofold by PND3 (*p* = 0.004), and then rose twofold by PND5 (*p* < 0.001). In OXYS rats, the density of mature neurons decreased from PND1 to PND3 (*p* = 0.005) and increased until PND5 (*p* < 0.001), although it remained lower than that of Wistar rats at this time point (*p* = 0.003).

In the PFC, the densities of neuroblasts and immature and mature neurons were affected only by age (F_4,48_ = 30.8, *p* < 0.0001; F_4,47_ = 83.3, *p* < 0.0001; and F_4,48_ = 16.5, *p* < 0.0001, respectively). In Wistar rats, the density of neuroblasts gradually increased from birth to PND3 with a large drop by PND5 (*p* = 0.002) followed by upregulation until PND7 (*p* = 0.003); in OXYS rats, the parameter significantly increased from PND0 to PND3 (*p* = 0.002), then decreased until PND5 (*p* < 0.0001) and rose by PND7 (*p* = 0.027). Of note, neuroblast density at PND7 was still significantly lower than that at PND3 in both rat strains (*p* = 0.015 for Wistar rats and *p* < 0.0001 for OXYS rats), indicating the completion of neuronal migration to the PFC. As for the density of immature neurons, this parameter significantly decreased from birth (for Wistar rats: *p* = 0.002 from PND0 to PND1; *p* = 0.007 from PND1 to PND3; *p* = 0.002 from PND3 to PND5; for OXYS rats: *p* = 0.003 from PND0 to PND1; *p* < 0.001 from PND3 to PND5), reached a minimum at PND5 and then increased until PND7 (*p* = 0.039 for Wistar rats and *p* = 0.048 for OXYS rats). Again, the density of immature neurons was lower at PND7 than at PND3 (*p* = 0.010 for both rat strains), indicating neuronal maturation. In the PFC of Wistar rats, the density of mature neurons remained constant from PND0 to PND1, decreased reaching a minimum at PND3 (*p* < 0.0001), increased until PND5 (*p* < 0.0001), and then dropped by PND7 (*p* < 0.001). In OXYS rats, the picture was the same with the exception of the peak of mature neuron density at PND5 (*p* = 0.009 from PND1 to PND3). For distributions of the cells of the neuronal lineage throughout the regions of the hippocampus and cortical layers, see Appendix A.

Altogether, our results show a decrease in the density of the neuronal-cell lineage in the PFC from birth to PND3 in both rat strains, and then from PND5 to PND7 only in Wistar rats (*p* < 0.001). The observed decrease in cell density might be caused by nucleus enlargement.

Consequently, the next stage of our study was the examination of neuronal nuclei sizes in the hippocampus and PFC (Table 2). In both brain regions, this parameter naturally increased with age (F_4,42_ = 76.0, *p* < 0.0001 for the hippocampus; F_4,40_ = 120.9, *p* < 0.0001 for the PFC). Indeed, in Wistar rats, we detected significant enlargement of neuronal nuclear size at all examined ages (for the hippocampus: *p* = 0.046 from PND0 to PND1, *p* < 0.001 from PND1 to PND3, *p* = 0.004 from PND3 to PND5, and *p* < 0.001 from PND5 to PND7; for the PFC: *p* = 0.004 from PND0 to PND1, *p* < 0.001 from PND1 to PND3, *p* < 0.001 from PND3 to PND5, and *p* = 0.010 from PND5 to PND7), whereas in OXYS rats, the parameter increased from PND0 to PND5 (for the hippocampus: *p* = 0.014 from PND0 to PND1, *p* = 0.003 from PND1 to PND3, and *p* = 0.047 from PND3 to PND5; for the PFC: *p* < 0.001 from PND0 to PND1, *p* < 0.001 from PND1 to PND3, and *p* = 0.002 from PND3 to PND5). It is important to emphasize that in the hippocampus, the size of neuronal nuclei was smaller in OXYS rats (F_1,42_ = 26.7, *p* < 0.0001).

### 3.4. Astrocytic Density in the Hippocampus and PFC of Neonatal OXYS and Wistar Pups

The density of astrocyte progenitors in the hippocampus was affected only by the age of the animals (F_4,80_ = 12.1, *p* < 0.0001). Indeed, from birth to PND1, the parameter increased two-fold in Wistar rats (*p* = 0.004) and four-fold in OXYS rats (*p* = 0.004), then diminished from PND3 to PND5 (*p* = 0.020 for Wistar rats and *p* = 0.007 for OXYS rats) and rose from PND5 to PND7 (*p* < 0.001 for Wistar rats and *p* = 0.038 for OXYS rats) in both rat strains (Figure 3a). Furthermore, the Newman–Keuls post hoc test suggested that OXYS rats were born with lower astrocyte progenitor cell density in the hippocampus (*p* = 0.029); however, the parameter reached a level similar to that of Wistar rats at PND1 (*p* = 0.78). As for astrocyte density in the hippocampus, this parameter was influenced by age (F_4,80_ = 4.4, *p* = 0.003), and there was an interaction between factors genotype and age (F_4,80_ = 5.4, *p* < 0.001). Indeed, we observed a difference in age-related dynamics of the hippocampal astrocyte density between OXYS and Wistar rats (Figure 3a). In Wistar rats, the astrocyte density remained comparatively unchanged from birth to PND3 and then went up by PND5 (*p* = 0.016), whereas in OXYS rats, the parameter increased from birth to PND1 (*p* = 0.003), reached a maximum at PND3, and decreased until PND5 (*p* = 0.031). Such dissimilar dynamics resulted in different astrocyte densities. Indeed, OXYS rats were born with lower astrocyte density in the hippocampus as compared to Wistar rats (*p* = 0.039). Then, the parameter reached the level of Wistar rats at PND1, was higher in OXYS rats at PND3 (*p* = 0.009), and lower at PND5 and PND7 (*p* = 0.048 and *p* = 0.035, respectively). For distributions of GFAP-positive cells throughout the regions of the hippocampus, see Figure 3c.

In the PFC, the density of astrocyte progenitors was affected only by age (F_4,80_ = 9.8, *p* < 0.0001; Figure 3b). In Wistar rats, the parameter increased from birth to PND1 (*p* < 0.001) and then from PND5 to PND7 (*p* = 0.003). OXYS rats were born with a lower density of astrocyte progenitors in comparison with Wistar rats (*p* = 0.033); the parameter increased by PND1 (*p* = 0.037), reaching the level of Wistar rats and then decreased from PND3 to PND5 (*p* = 0.001). The density of astrocytes (Figure 3b) was influenced by age (F_4,80_ = 12.1, *p* < 0.0001) and genotype (F_1,80_ = 7.0, *p* = 0.010), and there was an interaction between the two factors (F_4,80_ = 7.4, *p* < 0.0001). In Wistar rats, the density of astrocytes increased from birth to PND1 (*p* = 0.007) and then from PND5 to PND7 (*p* = 0.006). OXYS rats were born with a lower density of astrocytes relative to Wistar rats (*p* = 0.016). The parameter increased by PND1 (*p* = 0.004), but remained more than 1.5-fold lower than that of Wistar rats (without significance). The density of astrocytes continued to rise until PND3 in OXYS rats (*p* = 0.014), reaching the level of Wistar rats at this stage. We did not detect an increase in astrocyte density from PND5 to PND7 in OXYS rats; this phenomenon resulted in decreased astrocyte density in the PFC at PND7 as compared to Wistar rats (*p* = 0.004). For distributions of GFAP-positive cells throughout the cortical layers, see Figure 3d.

For distributions of astrocytes and their progenitors throughout the regions of the hippocampus and cortical layers, see Appendix A.

Next, we counted radial glial cells’ processes in the PFC (Table 3 and Figure 3e). Factorial ANOVA suggested that the number of radial glial-cell processes was affected by age (F_4,80_ = 19.0, *p* < 0.0001) and not by genotype (F_1,80_ = 2.3, *p* = 0.13); however, there was an interaction between the factors (F_4,80_ = 5.1, *p* < 0.001). Indeed, we revealed a difference in age-related dynamics of the number of radial glial-cell processes: in Wistar rats, the parameter reached a maximum at PND1, then decreased six-fold by PND3 (*p* = 0.002) and continued to go down with age reaching zero at PND7, whereas in OXYS rats, the parameter did not significantly change from birth to PND3, thereby revealing its tendency to be lower at PND0 and PND1 (*p* = 0.054 and *p* = 0.058, respectively) and significantly higher at PND3 (*p* = 0.010) compared to Wistar rats. After that, the number of radial glial-cell processes in OXYS rats decreased from PND3 to PND5 (*p* = 0.002).

A representative image of immunohistochemical staining for astrocytes is presented in Figure 3f.

### 3.5. The Density of Microglia in the Hippocampus and PFC of Neonatal OXYS and Wistar Pups

We showed that in the hippocampus, the total density of microglia (Figure 4a) increased from birth to PND7 (F_4,80_ = 20.2, *p* < 0.0001) and was lower in OXYS rats (F_1,80_ = 49.1, *p* < 0.0001). The densities of resting and activated microglia (Figure 4a) were affected by the genotype (F_1,80_ = 47.5, *p* < 0.0001 for resting microglia; F_1,80_ = 15.4, *p* < 0.001 for activated microglia) and age (F_4,80_ = 63.6, *p* < 0.0001 for resting microglia; F_4,80_ = 2.5, *p* = 0.046 for activated microglia); moreover, for resting-microglia density, there was an interaction between the factors (F_4,80_ = 4.2, *p* = 0.004). At birth, total microglial density in the hippocampus of OXYS rats was lower as compared to Wistar rats (*p* < 0.001), due to lowered density of both resting (*p* < 0.0001) and activated microglia (*p* = 0.022). At PND1, the total density of microglia in OXYS rats was lower as compared to Wistar rats (*p* = 0.014) because of lowered density of *activated* microglia (*p* = 0.021 PND1). On the contrary, at PND3, PND5, and PND7, the total density of microglia was lower (*p* = 0.045 for PND3; *p* = 0.016 for PND5; *p* < 0.001 for PND7) due to lowered density of resting microglia in OXYS rats (*p* = 0.003 for PND3; *p* = 0.010 for PND5; *p* < 0.001 for PND7). For distributions of microglia throughout the regions of the hippocampus, see Figure 4c.

In the PFC, total microglial density (Figure 4b) was influenced by age (F_4,80_ = 31.6, *p* < 0.0001) and was lower in OXYS rats (F_1,80_ = 18.9, *p* < 0.0001). The Newman–Keuls post hoc test suggested that in Wistar rats, microglial density did not significantly change from birth to PND5, and then increased by PND7 (*p* < 0.0001). In OXYS rats, the parameter increased from birth to PND1 (*p* = 0.007) and then from PND5 to PND7 (*p* < 0.0001). The density of resting microglia was affected by the genotype and age (F_1,80_ = 29.2, *p* < 0.0001 and F_4,80_ = 67.6, *p* < 0.0001, respectively), whereas the density of activated microglia was influenced only by age (F_4,80_ = 14.6, *p* < 0.0001). We did not find any differences in the total microglial density between OXYS and Wistar rats at PND0 and PND1, and detected only an insignificant decrease in total microglial density in the OXYS PFC at PND3 (*p* = 0.064). By contrast, at PND5 and PND7, the density of total microglia was significantly lower in OXYS rats (*p* < 0.001 for PND5 and PND7) due to diminished density of resting microglia (*p* < 0.001 for PND5 and *p* = 0.008 for PND7). For distributions of microglia throughout the cortical layers, see Figure 3d.

For distributions of the resting and activated microglia throughout the regions of the hippocampus and cortical layers, see Appendix A.

Then, we measured the area of microglial cells in the molecular layer of hippocampal DG, CA1, and CA3 regions and in cortical layer I because these layers had the highest microglial density (Table 4 and Figure 4e). ANOVA revealed that in the hippocampus, the area of microglia was affected by age (F_4,80_ = 11.0, *p* < 0.0001) and was smaller in OXYS rats (F_1,80_ = 23.6, *p* < 0.0001). Age-related alterations of this parameter were similar between Wistar and OXYS rats. We observed that microglial area enlarged from birth to PND1 (*p* = 0.002 for Wistar rats; *p* < 0.001 for OXYS rats), remained large until PND5, and then decreased by PND7 (an insignificant trend, *p* = 0.068 for Wistar rats; *p* = 0.012 for OXYS rats). Furthermore, the microglial area in the hippocampus was smaller in OXYS rats at PND0, PND1, and PND7 (*p* < 0.001, *p* = 0.009, and *p* = 0.017, respectively). As for the microglial area in the PFC, it was affected by age (F_4,80_ = 5.4, *p* < 0.001); in addition, there was a weak insignificant effect of the genotype (F_1,80_ = 3.2, *p* = 0.080) on the parameter. In the PFC, we saw similar age-related dynamics of microglial density between OXYS and Wistar rats. Indeed, this density rose from birth to PND1 in both rat strains (*p* = 0.003 for Wistar rats; *p* = 0.042 for OXYS rats), then declined by PND3 (*p* = 0.019 for Wistar rats; not significant for OXYS rats). Further age-related alterations of microglial area were insignificant in both rat strains: the parameter increased until PND5 and then diminished by PND7. We did not note any inter-strain differences in microglial area in the PFC.

After that, we estimated the percentage of pyknotic nuclei that were phagocytosed by microglia (Table 5 and Figure 4f). This parameter reflects functional consistency of microglia. We showed that in the hippocampus and PFC, the percentage of pyknotic nuclei phagocytosed by microglia increased with age (F_4,80_ = 34.6, *p* < 0.0001 for the hippocampus; F_4,80_ = 28.2, *p* < 0.0001 for the PFC) and was lower in OXYS rats (F_1,80_ = 23.3, *p* < 0.0001 for the hippocampus; F_1,80_ = 21.4, *p* < 0.0001 for the PFC). There was an interaction between factors age and genotype (F_4,80_ = 3.1, *p* = 0.019 for the hippocampus; F_4,80_ = 4.6, *p* = 0.002 for the PFC).

A representative image of immunohistochemical staining for resting and activated microglia is presented in Figure 4g.

### 3.6. Apoptosis in the Hippocampus and PFC of Neonatal OXYS and Wistar Pups

These data are presented in Table 6. In the hippocampus, the number of apoptotic cells was influenced by age (F_4,77_ = 59.3, *p* < 0.0001) and genotype (F_1,77_ = 33.4, *p* < 0.0001), and there was an interaction between the factors (F_4,77_ = 9.0, *p* < 0.0001). Indeed, we noted that at birth, the number of apoptotic cells was maximal in both rat strains and decreased more than twofold by PND1 (*p* = 0.005 for Wistar rats and *p* < 0.001 for OXYS rats). It is worth mentioning that the number of apoptotic cells was almost twofold higher in OXYS rats than in Wistar rats on PND0 (*p* = 0.007) and PND1 (*p* = 0.019). Then, we documented various changes of apoptosis in the hippocampus. In Wistar rats, apoptosis slightly decreased from PND1 to PND7, whereas in OXYS rats, it intensified until PND3 (*p* = 0.016), becoming more active as compared to Wistar rats (*p* < 0.0001), declined more than threefold by PND5 (*p* < 0.0001), and then continued to decrease until PND7 (*p* = 0.008).

In the PFC, the number of apoptotic cells was affected by age (F_4,67_ = 24.5, *p* < 0.0001) and genotype (F_1,67_ = 30.7, *p* = 0.002) of the rats. The number of apoptotic cells did not significantly change from PND0 to PND5 and diminished by PND7 in the PFC of both rat strains. The downregulation was fourfold in Wistar rats (*p* < 0.001) and threefold in OXYS rats (*p* < 0.001). The number of apoptotic cells at PND3 was greater in OXYS rats than in Wistar rats (*p* = 0.034).

## 4. Discussion

In the present study, we investigated the features of development of the hippocampus and PFC in senescence-accelerated OXYS rats during the first postnatal week. We demonstrated a disturbance in astroglial support of these structures as well as a microglial deficiency and higher intensity of apoptosis concurrently with a relatively unchanged neuronal population and hippocampal neurogenesis during a critical period for the formation of a network between these brain structures.

The brain-to-body weight ratio is a parameter that allows us to draw general conclusions about the state of the brain. Here, we showed a higher brain-to-body weight ratio in OXYS rats at PND5 because of decreased body weight at this age. We can hypothesize that the reduced fitness observed, even in young OXYS rats, leads to decreased body weight and is linked with a diet-related cardiovascular disorder late in life [27], which may be one of the potential risk factors for the development of AD signs [28].

Previously, we have documented signs of a delayed peak of neurogenesis in the hippocampal DG in OXYS rats at PND10 [23]. Here, we report the additional peak of ANP density in the DG of OXYS rats, which was registered on PND3 and coincided with higher density of neuroblasts in the granular layer of the DG, increased astrocytic density in the DG, and with a peak of apoptosis that always accompanies neurogenesis [25]. Taken together, these observations are suggestive of an additional peak of neurogenesis in the DG on PND3 in OXYS rats. Besides, we found that the peak of proliferating cells in the DG corresponds to PND1 in both strains. Nonetheless, we can speculate that these cells give rise to different cell lineages in OXYS rats as compared to Wistar rats. Indeed, in Wistar rats, there was an increase in neuroblast density in the hippocampus at PND3, which may point to the differentiation of the proliferating cells into a neuronal lineage. At the same time, in the hippocampus of OXYS rats, we noted an increase in astrocyte numbers at PND3 while the density of their progenitors remained unchanged, suggesting that the proliferating cells can differentiate into the glial lineage.

From an electrophysiological point of view, CA1 pyramidal neurons in the rat hippocampus undergo considerable alterations in their excitable membrane properties during the first month of postnatal development, and these changes (specifically the amplitude and duration of an action potential) are more pronounced in the first postnatal week [29]. In the hippocampal CA1 region at birth in OXYS rats, we noticed a higher density of neuroblasts and of immature neurons, potentially because of a delay of prenatal neuronal maturation. Nonetheless, already at PND1, the inter-strain differences disappeared. The observed decrease in mature-neuron density by PND3 in both rat strains may be related to the generation and maturation of synapses. Indeed, in the rat hippocampus, the formation and maturation of synapses increase significantly starting from the middle of the first postnatal week [30]. It should be mentioned that at PND5, there was lower mature neuron density in the pyramidal layer of the CA1 region and in the molecular layer of the CA3 region of OXYS rats. Cell bodies of GABAergic neurons are located in the molecular layers of hippocampal CA1 and CA3 regions [31]. Given the specific excitatory role of GABAergic neurons during the first week of rodent development (for review see [32]), their deficiency may have detrimental consequences for brain function.

It is widely accepted that in rodents, neuron migration to the cortex and the formation of cortical layers continue after birth [33]. The end of neuronal migration marks the beginning of a period of intense neuronal growth and synaptic formation in the neocortex, which starts in the middle of the first postnatal week in rats [34]. The largest changes in PFC neurons are seen during the first 10 days after birth; however, adultlike properties are not acquired until the end of the third week [35]. In the present work, we documented the completion of neuronal migration to the PFC at PND3 with subsequent intensive neuronal maturation until PND5. Although in Wistar rats the peak of the number of radial glial processes in the PFC was seen on PND1 and decreased until PND3, in OXYS rats the peak occurred at PND3. Additionally, in Wistar rats neuroblast density increased monotonically from birth to PND3, whereas in OXYS rats the greatest increase in this parameter took place from PND1 to PND3. Altogether, these results imply a delay in neuronal migration to the PFC in OXYS rats.

In this study, we noted a decrease in neuronal density in the hippocampus and PFC of both rat strains from birth to PND7. Our findings are in line with the data of Sellinger and coworkers [36], who demonstrated that neuronal density in the PFC of male rats diminishes from PND2 to PND6 and then increases by PND8. Sellinger and coworkers have linked the decrease in neuron density in the PFC throughout the early postnatal period with extensive dendritic growth and synapse formation [36]. We can speculate that the same is the case for the PFC of Wistar and OXYS rats. On the other hand, the decrease in the density of the neuronal cell lineage by PND7 may be at least in part explained by the observed significant increase in nuclear size by that age.

Multiple in vitro and in vivo studies suggest that astrocytes are more vulnerable to cellular stress than neurons are [37,38]. In the present work, we demonstrated that astrocyte density changes in the hippocampus and PFC of OXYS rats during the first week of life much more greatly than the density of neurons does. Indeed, OXYS pups were born with a lower density of astrocytes and their progenitors both in the hippocampus and PFC. Nevertheless, by PND1, these differences disappeared, and at PND3, astrocyte density was even higher in the hippocampus and layer I of the PFC known as the layer containing the majority of PFC astrocytes [39]. The upregulation of astrocyte density in the hippocampus was due to higher astrocyte density in the DG, which may be related to the additional peak of neurogenesis in the DG of OXYS rats at PND3. On the other hand, the rise of astrocyte density was temporary; for instance, at PND5, the density of astrocytes was more than 1.5-fold lower in the hippocampus of OXYS rats. It has been proven by Zhou and colleagues [40] that astrocytes from the hippocampus of mature rats are electrophysiologically passive, whereas astrocytes from the hippocampus of newborn rats (PND1–PND3) have voltage-gated outward K^+^ currents and inward Na^+^ currents. Astrocyte maturation—meaning the emergence of predominant leak-type K^+^ currents—starts on PND4 and proceeds during the first 3 postnatal weeks. Moreover, it is immature astrocytes that express thrombospondins promoting synaptogenesis [41]. Therefore, astrocyte support in the 3 weeks after birth, especially during the first half of the first postnatal week, is of utmost importance for synaptogenesis and accordingly for neuronal plasticity in an adult brain.

There are two waves of apoptosis during brain development: the first is prenatal and intended for elimination of proliferating cells to give rise to a pool of neuronal stem cells; the second wave is postnatal and eliminates migrating and differentiating neurons that are not able to form appropriate neuronal circuits [42]. Previously, we have reported a delay in the postnatal wave of apoptosis in the DG of OXYS rats [22]. In the present work, we demonstrated that at birth, the number of apoptotic cells in the hippocampus of OXYS rats is higher than that in Wistar rats. Despite the decreased density of microglial cells (both resting and activated) and smaller microglial area in the hippocampus of OXYS rats at birth, we did not observe any differences in the percentage of pyknotic nuclei phagocytosed by microglia between OXYS and Wistar rats. This finding could be due to a greater number of apoptotic cells observed in the *CA1* and *CA3* regions of the hippocampus, whereas a decrease in the density of activated microglia took place in the *DG* of OXYS rats. Thus, the activity of all microglial cells was directed toward phagocytosis of pyknotic nuclei as a consequence of cell death. It is well known that during development, microglia play a substantial part in the phagocytosis of dying cells as well as in synaptic pruning [14]. If the majority of microglial cells were directed to eliminate the detritus of apoptotic cells in the hippocampus of OXYS rats, then a lesser proportion of microglia was involved in synaptic pruning, thereby potentially altering the formation of synapses. As a consequence of higher apoptosis intensity and lower microglial density, there was a significantly lower percentage of the pyknotic nuclei phagocytosed by microglial cells in the hippocampus of OXYS rats. On the other hand, we detected pyknotic nuclei covered by astrocyte processes in the hippocampus of OXYS rats at PND3. We theorized that astrocytes may partly take over the role of phagocytic cells and clear the cell detritus. Therefore, the increased density of astrocytes in the hippocampus of OXYS rats at this age may be intended not only for neuronal support and neurogenesis but also for phagocytosis to complement microglial function. Taken together, these results may reflect a delay in the prenatal wave of apoptosis and subsequent insufficiency of microglial phagocytic function in the hippocampus of OXYS rats on the first days of life. Regarding the PFC, the additional peak of apoptosis in OXYS rats at PND3 at the end of neuronal migration to the cortex may arise to eliminate improperly migrating neurons.

To summarize the new findings, we detected an additional peak of DG neurogenesis that coincides in time with the peak of apoptosis in the hippocampus and PFC of OXYS pups. Furthermore, we showed delayed migration of neurons to the PFC as well as disturbances in astrocytic and microglial support of the PFC and hippocampus during the first postnatal week. These developmental events may have long-lasting consequences, giving rise to the neurodegenerative process observed late in life in OXYS rats. For instance, altered neurogenic-niche formation may be the cause of the previously reported alteration of the neurogenic-niche microenvironment and depletion of neuronal progenitors at an advanced age [22]. A deficiency of astrocytic support and previously shown altered formation of mossy fibers [23] may result in inappropriate synaptic formation and lead to the synaptic dysfunction observed in the hippocampus of OXYS rats late in life [19,21].

It is known that in rodents, the hippocampus and PFC are among the regions that are most sensitive to early life experiences [43,44]. Nevertheless, the outcome of damage to the brain depends upon the developmental phase during which the lesion occurs: early brain damage may cause disturbances of neurodevelopmental programs and may thus induce behavioral deficits in adults [45]. Indeed, prenatal stress-induced anxiety and depression-like behavior become more pronounced not only in early life development but also at an old age, as there are the two most vulnerable periods in life [46]. Hoeijmakers and coworkers [47] have revealed that exposure of APP/PS1 transgenic mice to stress early in life exacerbates Aβ plaque load and alters the immune response to Aβ neuropathology in the hippocampus of aged animals. In addition, the cerebral distribution of the AD pathology matches the pattern of brain regions that mature later in childhood and adolescence and retain their plastic capacities in adulthood, namely the hippocampus, PFC, and limbic cortex [4]. Thus, our data are in line with the studies cited above and are suggestive of a “two-hit hypothesis of AD” in the context of the “developmental origins of behavior, health and disease” theory [48]. The alterations of early hippocampal and PFC development in OXYS rats may be regarded as the “first hit,” which can be compensated for in young-to-adult animals [23]; however, an unknown second hit later in life may drive the development and progression of AD signs. Additional research is needed to test this hypothesis.

## Figures and Tables

**Figure 1 biomedicines-09-00823-f001:**
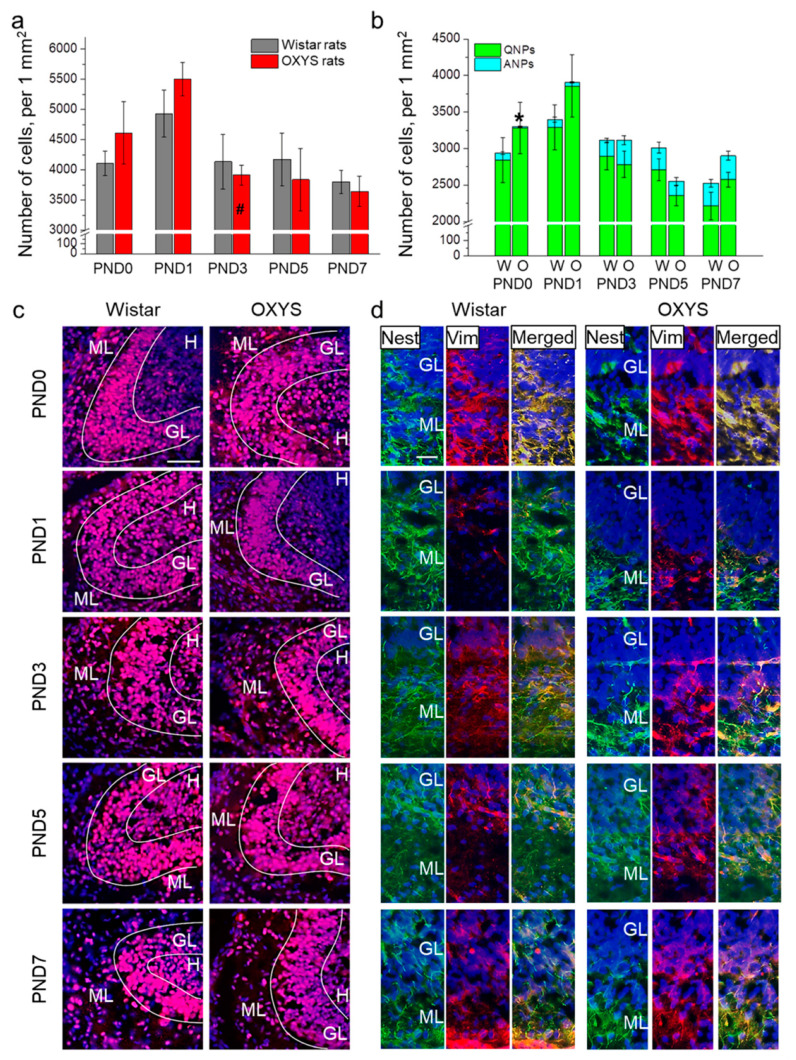
The density of proliferating cells and neuronal progenitors in the DG of OXYS and Wistar rats during the first postnatal week. (**a**) We did not observe any inter-strain differences in the density of proliferating cells in the DG; the only age-related difference was the decrease in proliferating cell density in OXYS rats from PND1 to PND3. (**b**) OXYS rats were born with a lower density of ANPs in the DG; nevertheless, by PND1 this difference had disappeared. (**c**) Immunohistochemical staining of the DG with antibodies against Ki-67 (red); the scale bar is 50 µm. (**d**) Immunohistochemical staining of the DG with antibodies against nestin (green) and vimentin (red); the scale bar is 20 µm. DAPI (blue) indicates cell nuclei. W: Wistar, O: OXYS, GL: granular layer of DG, ML: molecular layer of DG, H: hilus, Nest: nestin, Vim: vimentin. The data (**a**,**b**) are presented as mean ± SEM; * *p* < 0.05 for differences between the strains; ^#^
*p* < 0.05 for differences from a previous time point.

**Figure 2 biomedicines-09-00823-f002:**
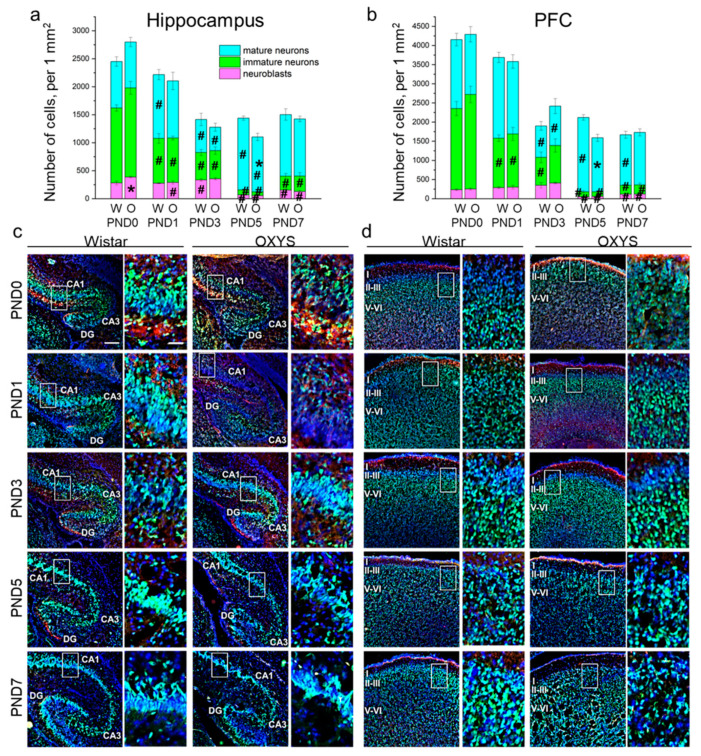
The density of neuroblasts and immature and mature neurons in the hippocampus (**a**) and PFC (**b**) of OXYS and Wistar rats during the first postnatal week. Microphotographs of the hippocampus (**c**) and PFC (**d**) stained by antibodies against NeuN (green) and DCX (red); DAPI indicates cell nuclei. Scale bars are 200 μm and 50 μm (within the frame). W: Wistar; O: OXYS; I: cortical layer I; II-III: cortical layers II–III; V-VI: cortical layers V–VI. The data (**a**,**b**) are presented as mean ± SEM; * *p* < 0.05 for differences between the strains; ^#^
*p* < 0.05 for differences from a previous time point.

**Figure 3 biomedicines-09-00823-f003:**
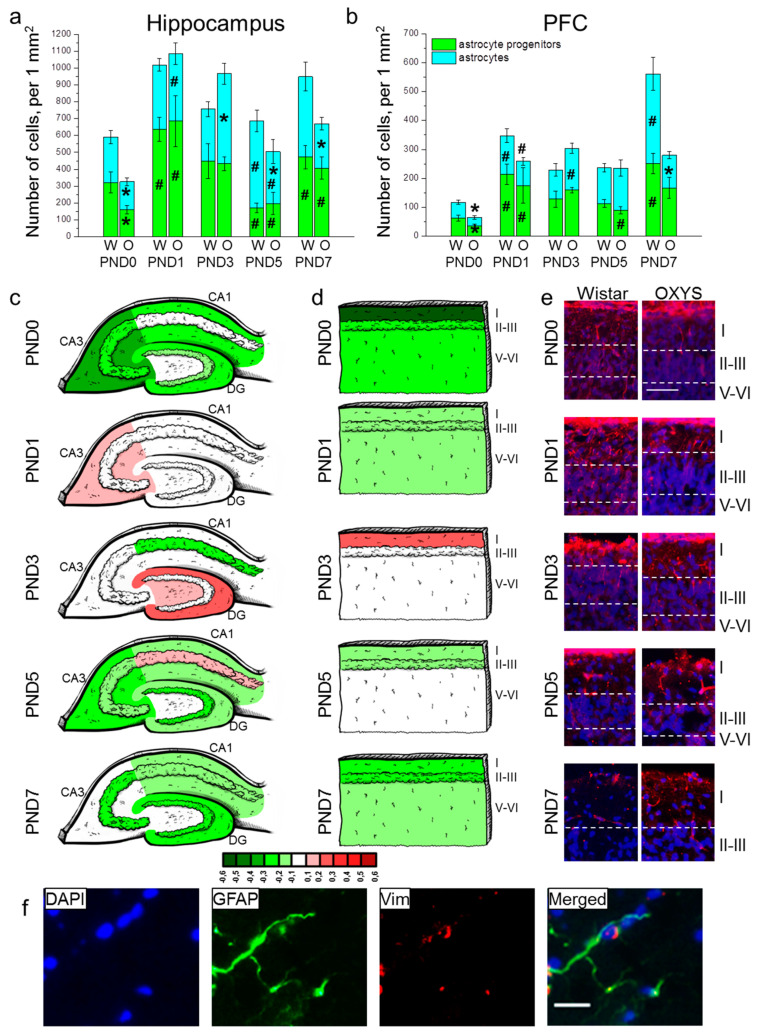
The density of astrocytes and their progenitors in the hippocampus (**a**) and PFC (**b**) of OXYS and Wistar rats during the first postnatal week. Changes in GFAP^+^ cell density in the regions of hippocampus (**c**) and layers of PFC (**d**) of OXYS rats compared to Wistar rats are presented schematically with GFAP^+^ cell density in each region and layer coded as heatmap with *lg* scale from −0.6 to 0.6. (**e**) Vimentin (red)-positive processes of radial glia in the PFC of Wistar and OXYS rats; the scale bar is 50 µm. (**f**) A representative image of GFAP (green)- and vimentin (red)-positive cells in the hippocampus of a Wistar rat at PND7; the scale bar is 20 µm. DAPI (blue) indicates cell nuclei (**e**,**f**). W: Wistar; O: OXYS; I: cortical layer I; II–III: cortical layers II–III; V-VI: cortical layers V–VI; Vim: vimentin. The data (**a**,**b**) are presented as mean ± SEM; * *p* < 0.05 for differences between the strains; ^#^
*p* < 0.05 for differences from a previous time point.

**Figure 4 biomedicines-09-00823-f004:**
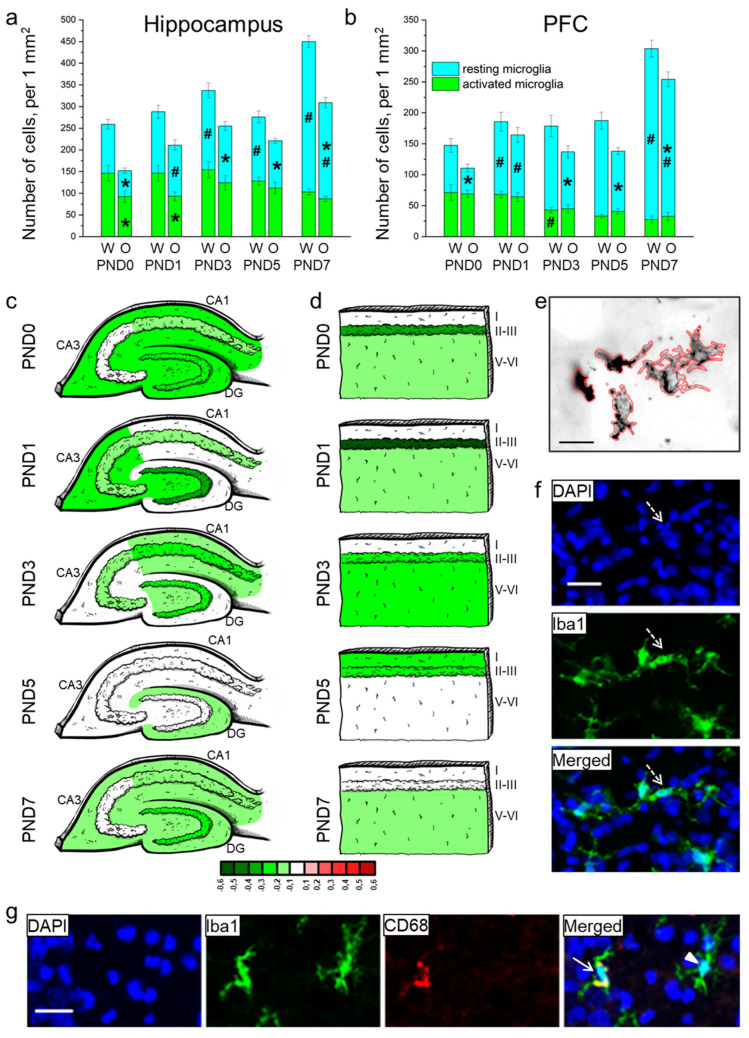
The density of microglia in the hippocampus (**a**) and PFC (**b**) of OXYS and Wistar rats during the first postnatal week. Changes in microglia density (meaning Iba1-positive cells) in regions of the hippocampus (**c**) and layers of the PFC (**d**) of OXYS rats compared to Wistar rats are presented schematically with Iba1^+^ cell density in each region and layer coded as a heatmap on a *lg* scale from −0.6 to 0.6. (**e**) The area of microglial cells was measured using ImageJ software. (**f**) A pyknotic nucleus (pointed out by a dotted arrow) covered by microglial cytoplasm. (**g**) A representative image of Iba1 (green)- and CD68 (red)-positive cells in the hippocampus of a Wistar rat at PND7; the arrow points to an activated microglial cell, and the arrowhead indicates a resting microglial cell. DAPI (blue) stains cell nuclei (**f**,**g**); the scale bar is 20 µm (**e**–**g**). W: Wistar; O: OXYS; I: cortical layer I; II-III: cortical layers II–III; and V-VI: cortical layers V–VI; solid arrow: an activated microglial cell; arrowhead: a resting microglial cell; dotted arrow: a pyknotic nucleus. The data (**a**,**b**) are presented as mean ± SEM; * *p* < 0.05 for differences between the strains; ^#^
*p* < 0.05 for differences from a previous time point.

**Table 1 biomedicines-09-00823-t001:** Body and brain weight and the brain-to-body weight ratio in OXYS and Wistar pups.

Time Point	Wistar Rats			OXYS Rats		
	Body Weight, g	Brain Weight, g	Brain-to-Body Weight Ratio, %	Body Weight, g	Brain Weight, g	Brain-to-Body Weight Ratio, %
PND1	7.08 ± 0.42	0.334 ± 0.008	4.84 ± 0.20	6.08 ± 0.31	0.258 ± 0.008 *	4.32 ± 0.17
PND3	9.07 ± 0.41 ^#^	0.443 ± 0.012 ^#^	4.94 ± 0.13	8.00 ± 0.41 ^#^	0.422 ± 0.008 ^#^	5.40 ± 0.24 ^#^
PND5	13.20 ± 0.55 ^#^	0.595 ± 0.016 ^#^	4.58 ± 0.16	10.25 ± 0.52 ^#,^*	0.568 ± 0.011 ^#^	5.66 ± 0.21 *
PND7	17.56 ± 0.59 ^#^	0.759 ± 0.016 ^#^	4.38 ± 0.13	15.23 ± 0.81 ^#,^*	0.714 ± 0.016 ^#^	4.84 ± 0.25 ^#^

* *p* < 0.05 for differences between the strains; ^#^
*p* < 0.05 for differences from a previous time point.

**Table 2 biomedicines-09-00823-t002:** The size of neuronal nuclei in the hippocampus and PFC of OXYS and Wistar pups (μm^2^).

Time Point	Wistar Rats		OXYS Rats	
	Hippocampus	PFC	Hippocampus	PFC
PND0	148.78 ± 5.05	90.27 ± 1.10	132.31 ± 1.70 *	89.22 ± 1.18
PND1	163.49 ± 2.95 ^#^	102.83 ± 2.45 ^#^	148.73 ± 4.45 ^#,^*	104.78 ± 1.85 ^#^
PND3	191.39 ± 3.14 ^#^	123.83 ± 1.69 ^#^	167.62 ± 2.31 ^#,^*	120.55 ± 2.21 ^#^
PND5	210.46 ± 3.39 ^#^	169.97 ± 6.30 ^#^	198.05 ± 13.20 ^#^	168.13 ± 10.61 ^#^
PND7	257.74 ± 6.63 ^#^	194.40 ± 3.19 ^#^	221.05 ± 3.59 *	187.26 ± 5.80

* *p* < 0.05 for differences between the strains; ^#^
*p* < 0.05 for differences from a previous time point.

**Table 3 biomedicines-09-00823-t003:** The number of radial glial cells’ processes in the PFC of OXYS and Wistar rats (counts per field of view).

Time Point	Wistar Rats	OXYS Rats
PND0	5.44 ± 0.82	3.44 ± 0.50
PND1	6.22 ± 1.41	2.89 ± 0.82
PND3	1.00 ± 0.29 ^#^	3.22 ± 0.70 *
PND5	0.89 ± 0.45	0.56 ± 0.24 ^#^
PND7	0	0.22 ± 0.15

* *p* < 0.05 for differences between the strains; ^#^
*p* < 0.05 for differences from a previous time point.

**Table 4 biomedicines-09-00823-t004:** The area of microglia in the hippocampus and PFC of OXYS and Wistar pups (µm^2^).

Time Point	Wistar Rats		OXYS Rats	
	Hippocampus	PFC	Hippocampus	PFC
PND0	162.74 ± 5.01	154.62 ± 10.41	125.42 ± 7.46 *	133.90 ± 17.50
PND1	195.50 ± 7.57 ^#^	222.71 ± 16.82 ^#^	166.21 ± 6.19 *^,#^	189.63 ± 18.21 ^#^
PND3	186.81 ± 9.84	161.20 ± 16.55 ^#^	177.68 ± 5.99	144.86 ± 18.69
PND5	190.80 ± 10.77	196.99 ± 17.11	171.77 ± 10.50	186.78 ± 17.71
PND7	164.59 ± 7.96	163.51 ± 12.50	134.28 ± 8.10 *^,#^	152.04 ± 16.14

* *p* < 0.05 for differences between the strains; ^#^
*p* < 0.05 for differences from a previous time point.

**Table 5 biomedicines-09-00823-t005:** The percentage of pyknotic nuclei phagocytosed by microglia in the hippocampus and PFC of OXYS and Wistar pups.

Time Point	Wistar Rats		OXYS Rats	
	Hippocampus	PFC	Hippocampus	PFC
PND0	29.79 ± 2.54	35.37 ± 3.35	34.73 ± 2.57	27.59 ± 3.97
PND1	43.21 ± 3.17 ^#^	50.00 ± 2.78 ^#^	29.78 ± 2.40 *	44.07 ± 7.75
PND3	59.04 ± 3.18 ^#^	79.44 ± 9.11 ^#^	44.31 ± 3.37 ^#,^*	29.44 ± 5.21 *
PND5	58.31 ± 5.53	64.66 ± 6.40	43.84 ± 4.47	49.26 ± 9.21
PND7	78.30 ± 3.29 ^#^	98.15 ± 1.85 ^#^	61.50 ± 3.96 ^#,^*	88.43 ± 5.80 ^#^

* *p* < 0.05 for differences between the strains; ^#^
*p* < 0.05 for differences from a previous time point.

**Table 6 biomedicines-09-00823-t006:** The number of apoptotic cells in the hippocampus and PFC of OXYS and Wistar pups.

Time Point	Wistar Rats		OXYS Rats	
	Hippocampus	PFC	Hippocampus	PFC
PND0	30.43 ± 3.92	3.00 ± 0.62	55.13 ± 6.24 *	5.25 ± 1.01
PND1	12.83 ± 2.98 ^#^	1.57 ± 0.43	22.86 ± 2.23 ^#,^*	3.29 ± 0.97
PND3	15.70 ± 2.36	2.13 ± 0.44	31.50 ± 2.15 ^#,^*	4.11 ± 0.70 *
PND5	8.50 ± 1.77	3.17 ± 0.48	9.00 ± 0.93 ^#^	3.33 ± 0.67
PND7	7.91 ± 0.95	0.75 ± 0.25 ^#^	6.44 ± 0.41 ^#^	1.05 ± 0.11 ^#^

* *p* < 0.05 for differences between the strains; ^#^
*p* < 0.05 for differences from a previous time point.

## Data Availability

Raw data are available from the corresponding author upon request.

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
