# Peer review of "Glia Not Neurons: Uncovering Brain Dysmaturation in a Rat Model of Alzheimer’s Disease"

_biomedicines, 2021, doi:10.3390/biomedicines9070823_

Round 1

Reviewer 1 Report

The manuscript by Rudnitskaya et al., is about the comparisons of cell density change during the first postnatal week between normal and Alzheimer model rats. The authors found a slight differences of densities of neurons, astrocytes, and microglia in some of the time points, especially around PND5-7. The method employed seems adequate. However, the reviewer feels that the authors tried to conclude too much from their observations: densities of cell types. The manuscript is very hard to read because of ineffective usage of figures and repeats of same statement throughout the manuscript. The paper must be rewritten to be more concise report of what they found.

  1. Title: What does the “key players in brain dysmaturation” mean? Authors suggested that glia is more important that neurons in neurodegenerative disease. However, what they showed is only the difference of densities. This is overstatement.
  2. Line 19: Show the clear evidence about “delayed migration of neurons to the prefrontal cortex”. Is the subtle difference of radial glia processes between strains enough to conclude delayed migration?
  3. Quality of images is good. However, detailed description of images are not provided, so it is quite difficult for readers to comprehend the meaning of images.
  4. Indicate exact p-values instead of using “p<0.05” etc., with exception for “p<0.001” (exact p-value is not necessary if p<0.001).
  5. Discussion is quite redundant. The most part (line 451-621) is repeat of results and can be deleted or shortened 

Author Response

The manuscript by Rudnitskaya et al., is about the comparisons of cell density change during the first postnatal week between normal and Alzheimer model rats. The authors found a slight differences of densities of neurons, astrocytes, and microglia in some of the time points, especially around PND5-7. The method employed seems adequate. However, the reviewer feels that the authors tried to conclude too much from their observations: densities of cell types. The manuscript is very hard to read because of ineffective usage of figures and repeats of same statement throughout the manuscript. The paper must be rewritten to be more concise report of what they found.

  1. Title: What does the “key players in brain dysmaturation” mean? Authors suggested that glia is more important that neurons in neurodegenerative disease. However, what they showed is only the difference of densities. This is overstatement.

We have changed the title of the article making it more appropriate for the obtained results.

  1. Line 19: Show the clear evidence about “delayed migration of neurons to the prefrontal cortex”. Is the subtle difference of radial glia processes between strains enough to conclude delayed migration?

This statement was based not only on the number of radial glia processes but also indirectly on the rate of changes of neuroblast density in the prefrontal cortex. Nevertheless, we agree with your comment. We have toned down this statement in the abstract.

  1. Quality of images is good. However, detailed description of images are not provided, so it is quite difficult for readers to comprehend the meaning of images.

Thanks for the comment; we have added detailed descriptions of the images.

  1. Indicate exact p-values instead of using “p<0.05” etc., with exception for “p<0.001” (exact p-value is not necessary if p<0.001).

We now indicate exact p values up to the third decimal place.

  1. Discussion is quite redundant. The most part (line 451-621) is repeat of results and can be deleted or shortened

We have shortened the Discussion section.

Reviewer 2 Report

In their work Rudnitskaya et al. were interested in the connection between developmental impairments of the CNS and the occurrence of neurodegenerative diseases later in life. They took advantage of an established OXYS experimental rat model of accelerated aging. The data presented show an additional peak of neurogenesis together with delayed migration of neurons in the prefrontal cortex. As a main finding the authors also describe impairments in the development of astrocytes and microglia in the hippocampus and prefrontal cortex in OXYS rats compared to control animals during the first postnatal week.

The experiments are thoroughly performed and the statistics are adequate. The description could be in parts a little more focused and shorter, also the discussion is very long. My main criticism is the lack of evidence of indeed a connection of developmental impairments with the occurrence of AD-like symptoms later on in the rat model. This is the main angle in the abstract and introduction, however, I do not find significant proof for this connection in the data presented.  In the discussion indeed the authors focus mostly on the developmental aspects and only in the very last part comment on the – of course very interesting – potential connection of impairments in network formation and glia function during the first postnatal week and a correlation with AD-like pathology later. In my opinion, this distribution is adequate (although the discussion in general is rather long) as there is no evidence for a connection and therefore it should only be discussed briefly. I therefore recommend a refocusing of the main part of the paper (including abstract and introduction) on the developmental impairments.

Furthermore my detailed points are:

  1. In Table 1, the authors summarized that brain weight and body weight differ slightly between Wistar and OXYS mice, and at PND5, the brain/body weight ratio is higher in OXYS mice. Although at this time point the brain weight does not show much difference from Wistar rats, indeed the body weight is lower which therefore alters the brain/body weight ratio. To what extent can this general phenotype of OXYS rats with potentially reduced fitness (indicated by the reduced body weight) be involved in CNS impairments and AD-like pathology later. Can this phenotype be associated with brain dysfunction and the onset of AD-like symptoms? This point should be discussed.
  2. Since the density of proliferating cells (Ki67+) is increased on PND1 in both Wistar and OXYS rats, can the authors discuss to which cell population these proliferating cells belong (neuronal or glial cells)?
  3. How did the authors distinguish resting and activated microglial populations? Was this done morphologically as CD68 is also expressed at low levels in resting microglia it is not a suitable marker to reveal purely activated microglial cells.
  4. Although microglia density and phagocytosis are reduced in OXYS rats compared to Wistar rats, the apoptosis rate is higher, by which mechanisms are apoptotic cells removed from the hippocampus and PFC of OXYS rats?
  5. The authors did not discuss the mechanisms how impairments in network development and glial support during the first postnatal days may be causally responsible for AD pathogenesis later (e.g. Aβ plaques formation).

Author Response

In their work Rudnitskaya et al. were interested in the connection between developmental impairments of the CNS and the occurrence of neurodegenerative diseases later in life. They took advantage of an established OXYS experimental rat model of accelerated aging. The data presented show an additional peak of neurogenesis together with delayed migration of neurons in the prefrontal cortex. As a main finding the authors also describe impairments in the development of astrocytes and microglia in the hippocampus and prefrontal cortex in OXYS rats compared to control animals during the first postnatal week.

The experiments are thoroughly performed and the statistics are adequate. The description could be in parts a little more focused and shorter, also the discussion is very long. My main criticism is the lack of evidence of indeed a connection of developmental impairments with the occurrence of AD-like symptoms later on in the rat model. This is the main angle in the abstract and introduction, however, I do not find significant proof for this connection in the data presented.  In the discussion indeed the authors focus mostly on the developmental aspects and only in the very last part comment on the – of course very interesting – potential connection of impairments in network formation and glia function during the first postnatal week and a correlation with AD-like pathology later. In my opinion, this distribution is adequate (although the discussion in general is rather long) as there is no evidence for a connection and therefore it should only be discussed briefly. I therefore recommend a refocusing of the main part of the paper (including abstract and introduction) on the developmental impairments.

Furthermore my detailed points are:

  1. In Table 1, the authors summarized that brain weight and body weight differ slightly between Wistar and OXYS mice, and at PND5, the brain/body weight ratio is higher in OXYS mice. Although at this time point the brain weight does not show much difference from Wistar rats, indeed the body weight is lower which therefore alters the brain/body weight ratio. To what extent can this general phenotype of OXYS rats with potentially reduced fitness (indicated by the reduced body weight) be involved in CNS impairments and AD-like pathology later. Can this phenotype be associated with brain dysfunction and the onset of AD-like symptoms? This point should be discussed.

Thank you very much for the insightful question. We have addressed it in the Discussion.

  1. Since the density of proliferating cells (Ki67+) is increased on PND1 in both Wistar and OXYS rats, can the authors discuss to which cell population these proliferating cells belong (neuronal or glial cells)?

Unfortunately, we have no direct evidence; thus, we can only hypothesize the possible fate of the proliferating cells on the basis of the obtained data. We inserted our hypothesis into the Discussion section.

  1. How did the authors distinguish resting and activated microglial populations? Was this done morphologically as CD68 is also expressed at low levels in resting microglia it is not a suitable marker to reveal purely activated microglial cells.

We appreciate these questions, yes, this is definitely the case. We determined the number of cells with distinct presence of the protein of interest, rather than the intensity of fluorescence. Therefore, if the protein was expressed weakly, we couldn’t detect it. Besides, there are reports in which authors distinguished Iba1+CD68- and Iba1+Cd68+ microglia by methods of cell counting. For example, Waller and colleagues (Waller et al., 2019) distinguished the following types of microglia: Iba1+Cd68- with a ramified profile and Iba1+CD68+ microglia with a phagocytic profile.

  1. Although microglia density and phagocytosis are reduced in OXYS rats compared to Wistar rats, the apoptosis rate is higher, by which mechanisms are apoptotic cells removed from the hippocampus and PFC of OXYS rats?

We hypothesized that one of the possible players is astrocytes, which phagocytosed the excessive cellular detritus. Indeed, we observed pyknotic nuclei surrounded by astrocyte processes in the hippocampus of OXYS rats. We now mention this finding in the Discussion section, but it seems not enough. Thus, we added a discussion of the phenomena.

  1. The authors did not discuss the mechanisms how impairments in network development and glial support during the first postnatal days may be causally responsible for AD pathogenesis later (e.g. Aβ plaques formation).

Thank you for noticing; we have addressed this issue in the Discussion section.

Round 2

Reviewer 1 Report

In general, authors' corrections are not sufficient to what I pointed. For example,

  1. P value style is not changed.
  2. In figures, authors did not describe what are the images mean. Instead, authors inserted long description for bar graphs. They can be moved in results.
  3. Discussion is still too long.

Author Response

1. P value style is not changed.

We provided the exact meaning of the p-value. We are greatly hope that we have done what you meant.

2. In figures, authors did not describe what are the images mean. Instead, authors inserted long description for bar graphs. They can be moved in results.

We are greatly hope that we have understood you correctly, we added the description of the heatmaps to Figures 3 and 4.

Discussion is still too long.

We shortened the Discussion section.

Reviewer 2 Report

  1. I still believe that in some parts of the manuscript the link between developmental defects and AD-like pathology is overemphasized as the authors cannot provide evidence to support this hypothesis. It should be made clear that they describe an early phenotype which might be linked to later phenotypes. So in line 78 of the introduction the phase this hypothesis (which is interesting and also their phenotype might in the future be linked to pathological alterations during aging) and then continue to with …to test this hypothesis in line 82. In my opinion they did not experimentally investigate the link between pathological alterations during development and aging but only describe the developmental phenotype. So it should be rephrased that future experiments might be able to link this developmental phenotype they uncovered to AD-pathology.
  2. In my opinion the figure legends are too long and should be more condensed still providing necessary information.
  3. The discussion is still too long and should be also more condensed.

Author Response

1. I still believe that in some parts of the manuscript the link between developmental defects and AD-like pathology is overemphasized as the authors cannot provide evidence to support this hypothesis. It should be made clear that they describe an early phenotype which might be linked to later phenotypes. So in line 78 of the introduction the phase this hypothesis (which is interesting and also their phenotype might in the future be linked to pathological alterations during aging) and then continue to with …to test this hypothesis in line 82. In my opinion they did not experimentally investigate the link between pathological alterations during development and aging but only describe the developmental phenotype. So it should be rephrased that future experiments might be able to link this developmental phenotype they uncovered to AD-pathology.

Thank you for the detailed comment. We tried to tone down the Introduction section making it more appropriate to the goal of the study.

2. In my opinion the figure legends are too long and should be more condensed still providing necessary information.

We deleted the excess text from figure legends.

3. The discussion is still too long and should be also more condensed.

We shortened the Discussion section.